# Evaluation of Morphological Changes in Retinal Vessels in Type 1 Diabetes Mellitus Patients with the Use of Adaptive Optics

**DOI:** 10.3390/biomedicines10081926

**Published:** 2022-08-09

**Authors:** Wojciech Matuszewski, Katarzyna Gontarz-Nowak, Joanna M. Harazny, Elżbieta Bandurska-Stankiewicz

**Affiliations:** 1Clinic of Endocrinology, Diabetology and Internal Medicine, Department of Internal Medicine, University of Warmia and Mazury, 10-082 Olsztyn, Poland; 2Department of Human Physiology and Pathophysiology, University of Warmia and Mazury, 10-082 Olsztyn, Poland; 3Department of Nephrology and Hypertension, University Hospital Erlangen, Friedrich Alexander University Erlangen Nuremberg (FAU), 91054 Erlangen, Germany

**Keywords:** diabetes mellitus, diabetic retinopathy, adaptive optics

## Abstract

Introduction. Diabetes mellitus contributes to the development of microvascular complications in the eye. Moreover, it affects multiple end organs, including brain damage, leading to premature death. The use of adaptive optics technique allows to perform non-invasive in vivo assessment of retinal vessels and to identify changes in arterioles about 100 μm in diameter. The retinal vasculature may be a model of the cerebral vessels both morphologically and functionally. **Aim**. To evaluate morphological parameters of retinal arterioles in patients with type 1 diabetes mellitus (DM1). **Material and methods.** The study included 22 DM1 patients (13 females) aged 43.00 ± 9.45 years with a mean diabetes duration of 22.55 ± 10.05 years, and 23 healthy volunteers (10 females) aged 41.09 ± 10.99 years. Blood pressure, BMI, waist circumference, and metabolic control markers of diabetes were measured in both groups. Vascular examinations were performed using an rtx1 adaptive optics retinal camera (Imagine Eyes, Orsay, France); the vessel wall thickness (WT), lumen diameter (LD), wall-to-lumen ratio (WLR), and vascular wall cross-sectional area (WCSA) were assessed. Statistical analysis was performed with the application of IMB SPSS version 23 software. **Results**. The DM1 group did not differ significantly in age, BMI, waist circumference, blood pressure, or axial length of the eye compared to the control group. Intraocular pressure (IOP) in both groups was normal, but in the DM1 group it was significantly higher. The DM1 group had significantly higher WT, WLR, and WCSA. These parameters correlated significantly with the duration of diabetes, but not with IOP. **Conclusions**. The presented study demonstrates the presence of significant morphological changes in retinal vessels in DM1 patients without previously diagnosed diabetic retinopathy. Similar changes may occur in the brain and may be early indicators of cardiovascular risk, but further investigation is required to confirm that.

## 1. Introduction

Diabetes mellitus (DM) has been known to mankind for over 3500 years. Its first clinical description was found in the Ebers Papyrus from 1530 BC, discovered in an Egyptian tomb in 1867, and another one in a papyrus discovered in 1899 in Dayr al-Barsha [1,2]. Currently, according to the WHO, DM is defined as a group of metabolic diseases with hyperglycaemia resulting from a defect in insulin secretion or action. Chronic hyperglycaemia causes damage, dysfunction, or insufficiency of various organs, especially the eyes, kidneys, nerves, the heart, and blood vessels [3]. The aetiological classification of diabetes distinguishes type 1 diabetes (DM1), which can be divided into autoimmune subtype, with the presence of antibodies: ICA, IA2, anti-GAD, ZNT8; and idiopathic subtype [4,5]. DM is non-infectious pandemic of the modern world, affecting 463 million adults worldwide, and it is predicted that the number of patients will increase to 700 million in 2045. DM1 accounts for 5–10% of all diabetes cases worldwide, the incidence of DM1 is 15 per 100,000 people [6,7]. DM leads to the development of macrovascular and microvascular complications which affect, among others, the organ of vision. The most serious of these complications is diabetic retinopathy (DR), defined as a highly specific neurovascular eye complication caused by diabetes. DR can concern one third of patients; it develops slowly, not giving any symptoms for a long time, and at the same time it accounts for 80% of vision loss in the diabetic population [8,9,10]. The term *retinitis diabetica* was introduced in 1885 by Eduard von Jaeger, and identifying the condition was possible because of the invention of the ophthalmoscope by Hermann von Helmholtz in 1851 [11,12]. Nowadays, in addition to direct ophthalmoscopy, fundoscopy, fluorescein angiography (FA), optical coherence tomography (OCT), ultrasonography (USG), or confocal microscopy are also used when diagnosing DR [13,14]. However, it is adaptive optics (AO) that causes the greatest excitement among researchers. Developed in 1953 by California-born astronomer Horace Babcock, it was first used to improve the performance of optical systems by reducing the effect of incoming wavefront distortion [15,16]. AO was initially used in astronomy and military technology, in telescopes and laser communication [17]. The first study with the use of AO in medicine was conducted by David Williams from Rochester, USA, director of Centre for Visual Science, who noticed a huge potential of this technique and built the first “telescope” to visualise the cones [18]. Today, AO is used in retinal imaging systems to reduce optical aberrations generated mainly by the lens and cornea. A device with AO uses a near-infrared wavefront distortion sensor, a deformable mirror or liquid crystal matrix to correct these aberrations in real time. Images of fundus structures with a resolution of 2–3 µm are obtained [19,20]. Thanks to the AO technique it is possible to non-invasively visualise: retinal vascularisation (vessels from 20 μm to more than 150 μm in diameter), photoreceptors (cones and rods), retinal nerve fibre layers, axons of ganglion cells, the retinal pigment epithelium, or lamina cribrosa [21,22,23]. AO makes it possible to detect lesions at a very early stage of development, undetectable with other techniques. Complementing the ophthalmologic examination with the opportunities offered by AO currently ensures the most accurate assessment of the eye in diabetic patients. The current study focuses on patients with DM1, but it should not be forgotten that AO helps in diagnosing hypertensive retinopathy, glaucoma, age-related macular degeneration, inflammation or dystrophy of retinal structures both in diabetic patients and in the general population [24,25,26,27,28]. Morphological studies of the retinal vascular system have great diagnostic potential. The eye is transparent and retinal resistance arterioles can be examined non-invasively. Supplying the 3rd afferent neuron of phototransduction, not only morphologically (outer blood–nerve barrier with tight junctions, surrounding of astrocytes and pericytes), or by common origin (with arteria carotis interna), but also functionally (myogenic and local autoregulation, neurovascular coupling, lack of autonomic innervation), these vessels are a proposed model of cerebral microcirculation, whose visualisation is not yet possible at this level [29,30]. Since cerebral microangiopathies in DM are a serious threat to patients’ lives, therefore, the study of retinal microangiopathies can indicate directions for the treatment of diabetic patients. The aim of the current study is to investigate with the use of AO parameters determining the morphology of retinal arterioles in patients with DM1 as compared to healthy individuals.

## 2. Materials and Methods

### 2.1. The Analysed Population

The study of retinal imaging with AO in patients with DM1 was conducted for a period of 12 months in 2021. The study group consisted of adult patients with DM1 diagnosed according to the WHO criteria who did not have DR as shown in fundus imaging [6]. All patients were treated with functional intensive insulin therapy in line with the current guidelines of the European Association Study of Diabetes (EASD) [5]. The study was conducted in accordance with the standards of good clinical practice and included patients under permanent care of a specialist diabetes clinic at the Clinic of Endocrinology, Diabetology and Internal Medicine of the University of Warmia and Mazury in Olsztyn. The study was approved (approval number 10/2010, date 25 March 2010) by the Bioethics Committee of the Faculty of Medicine, University of Warmia and Mazury in Olsztyn, Poland.

### 2.2. Metabolic Control Assessment Methods

Patients’ metabolic status was assessed on the basis of a questionnaire designed for the purpose for the study including information on: patient demographics, clinical history of diabetes and ocular history, and elements of physical examination. Hypertension was defined according to the European Society of Hypertension (ESH) and European Society of Cardiology (ESC) criteria, BMI was calculated by applying Quetelet’s equation, while the waist circumference measurement was interpreted according to the International Diabetes Federation (IDF) criteria [31,32,33]. In addition, the following laboratory markers were assessed: glycated haemoglobin (HbA1c), complete lipid profile, creatinine levels, glomerular filtration rate (GFR), and urine albumin/creatinine ratio (ACR) (Table 1).

### 2.3. Assessment Methods of the Eye

Examination of the fundus was performed by an experienced ophthalmologist with the use of direct ophthalmoscopy and colour two-field fundus photography (50 degree angle), including images of the optic nerve disc in the centre and the retinal macula in the centre, taken with a Topcon TRC NW8 fundus camera after pupil dilation with 1% Tropicamide. The fundus was assessed according to the International Clinical Classification System for Diabetic Retinopathy criteria: no diabetic retinopathy, non-proliferative diabetic retinopathy (NPDR), proliferative diabetic retinopathy (PDR), diabetic macular edema (DME) [34]. The eyeball axial length was determined using an optical biometer IOL Master 500 (Zeiss, Jena, Germany) and intraocular pressure was measured with a non-contact tonometer type Air-Puff TX-20 (Canon, Tokyo, Japan).

Vascular examinations were performed with the use of an rtx1 camera for human retinal examinations with adaptive optics (Imagine Eyes, Orsay, France) in line with the procedure recommended by the manufacturer. The retina was non-invasively studied in vivo at the cellular and microvascular level by assessing: vessel wall thickness (wall 1 + wall 2) (WT), lumen diameter (LD), and automatically calculating: wall-to-lumen ratio (WLR) and vessel wall cross-sectional area (WCSA). All measurements of retinal arterioles were made in the superior temporal quadrant and were performed in triplicate, calculating the arithmetic mean of the taken measurements. Patients were examined after 10 min of rest in a sitting position, after blood pressure measurement in the dark without pupil dilation in an air-conditioned room (in the temperature of 23 °C) by an observer with more than 25 years of experience of retinal vascular examination. The study was observer blinded. Blood pressure tests were performed using an Omron M3 blood pressure monitor (Omron, Kyoto, Japan).

### 2.4. Statistical Analysis

Statistical analysis was performed with the use of the IMB SPSS version 23 program (IBM, New York, NY, USA). The *p*-value of differences of unpaired median data with normal distribution was calculated with the use of the parametric *t*-Student test; and for unpaired measurements with non-normal distribution of results in at least one of the studied groups, the differences in median values were compared with the use of the non-parametric U-Mann Whitney test. The one-sample Kolmogorov–Smirnov test was used to determine the distributions of measurement values. Correlations of the studied parameters were performed with the age, gender, and BMI adjustments. The result of the analyses was considered statistically significant at *p* < 0.05.

## 3. Results

The study included 45 adults, with 22 DM1 patients (13 women) without DR features, aged 43.00 ± 9.45 years with a mean duration of diabetes of 22.55 ± 10.05 years, and 23 healthy volunteers (10 women) aged 41.09 ± 10.99 years (who were the control group). The group of patients with DM1 showed many similarities to the control group, differing statistically significantly only in higher systolic blood pressure and LDL cholesterol levels. The DM1 patient group had an elevated HbA1c value compared with the recommended range (Table 2).

The DM1 group did not differ significantly in the eyeball axial length, compared to the control group: right 23.6 ± 1.1 vs. 23.7 ± 1.0; left 23.5 ± 1.12 vs. 23.8 ± 1.2, respectively (*p* = 0.62). IOP in both groups was normal, but significantly higher in the DM1 group, compared to the control group, 16.89 ± 3.38 vs. 14.07 ± 1.83mmHg, respectively (*p* = 0.007) (Figure 1).

By analysing morphological parameters, it can be seen that WLR and WT in the DM1 group are statistically significantly higher compared to the control group (Figure 2). Furthermore, tendentially higher WCSA values were found in the DM1 group (Table 3).

In the DM1 group, WT, WLR, and WCSA correlated significantly (*p* < 0.05) with the diabetes duration but not with IOP. The correlations of WRL with the diabetes duration are shown in the figure below (Figure 3).

Multivariate analysis showed a positive correlation between WT, WLR, WCSA, and diabetes duration, considering age, gender, and BMI (Table 4).

## 4. Discussion

The current research demonstrates the presence of significant morphological changes in retinal vessels in patients with DM1. The changes we observed are not specific only to DM1, being a result of hyperglycaemia, they can also occur in other types of diabetes. Significantly higher values of WT, WLR, and tendentially higher WCSA values were found in the group of patients with DM1 compared to the healthy control group. The two analysed groups did not differ significantly in terms of age, BMI, waist circumference, blood pressure, or the eyeball axial length; thus, detecting the above changes is all the more valuable because the study encompassed DM1 patients without previously diagnosed retinal pathologies. Furthermore, it revealed a positive correlation between WLR and diabetes duration, and in multivariate analysis it showed a positive correlation between WLR, WT, WCSA, and diabetes duration with age, gender, and BMI adjustment. 

The first non-invasive study of WLR in retinal vessels was conducted in 2007 by the Schmieder group from Erlangen, Germany, in a group of patients with hypertension. Using Scanning Laser Doppler Flowmetry and a dedicated software to calculate WLR from reflexion and perfusion images of retinal arterioles, they showed that WLR of retinal arterioles increases with the age of patients. Moreover, WLR is significantly higher in the group of patients with ineffectively treated hypertension and after a stroke compared to the control group of the same age range [35]. Results of these studies aroused interest of researchers dealing with vascular remodelling in hypertension but also in diabetes, which facilitated the development of technology for non-invasive morphological measurements of retinal arterioles and the construction of rtx1.

A study of DM1 patients with the use of AO was already conducted by Lombardo in 2013. Analysis of only eight DM1 patients with a mean age of 39.50 ± 5.96 years with NPDR without DME revealed a smaller mean vessel lumen calibre compared to healthy subjects, respectively (6.27 ± 1.63 μm vs. 7.31 ± 1.59 μm, *p* = 0.002). The patients studied by Lombardo, as in the current study, did not differ in the eyeball axial length and had poor metabolic control of diabetes expressed by a mean HbA1c percentage of 7.50 ± 0.73% [36].

One year later, a team of researchers from Indiana, USA, used an adaptive optics scanning laser ophthalmoscope (AOSLO) to examine 6 DM1 patients aged 44 ± 6.6 years with already diagnosed mild NPDR. The treatment group was characterised by poor diabetes compensation with HbA1c >8.0% (8.1–9.2%) and a long duration of diabetes, on average >17 years. The control group consisted of healthy volunteers. DM1 patients had a larger capillary diameter compared to healthy controls, 8.2 ± 1.1μm vs. 6.1 ± 0.75 μm, and as in the current study, a larger WLR of retinal arterioles, 1.1 ± 0.07 vs. 0.48 ± 0.28 (*p* < 0.05). The researchers suggested that used DR classifications inaccurately describe and divide particular NPDR subtypes [37]. In 2019, an interesting publication was presented by Zaleska-Żmijewska et al., on the use of the AO technique in patients with diabetes. The study encompassed 36 patients, including 22 with mild NPDR and 14 with moderate NPDR, while the control group consisted of healthy subjects. It was shown that the mean vessel lumen and the total retinal artery diameter were not statistically significantly different in the group with DR and the control group (*p* = 0.580). However, the parameters WT, WLR, and WCSA were statistically significantly higher in the group with DM, which also confirms findings obtained in the current study. Furthermore, the authors showed a positive correlation of WLR and WCSA with the advancement of DR. When analysing metabolic markers, WCSA correlated positively with BMI, with the highest value (6529 μm^2^) in the group with BMI > 30.0 (*p* = 0.189). It was concluded that WLR is the best marker confirming retinal vascular remodelling, and that the detected vascular changes may be a strong risk factor for cardiovascular changes [38]. In another study, published in 2021, Ueno et al., reported the results of a large clinical trial on patients with DM2 performed with the used the AO technique, in which vascular morphological changes and blood flow were analysed. In the study, as in the present work, 47 patients without DR were included, as well as 36 with mild NPDR, 22 with severe NPDR, 32 with PDR, while the control group consisted of 24 volunteers without diabetes. WLR was statistically significantly higher in the group with PDR compared to the other groups. WLR correlated positively with the mean blur rate (MBR)-vessel (r = −0.337, *p* < 0.001), advancement of DR (r = 0.643, *p* < 0.001), systolic blood pressure (r = 0.166, *p* < 0.037), and presence of systemic hypertension (r = 0.443, *p* < 0.001). As in the current study, a statistically significantly higher WLR was associated with the duration of diabetes (r = 0.191, *p* = 0.042). Another analogy concerned WT, which, as in the current study, was statistically significantly higher in the diabetic group compared with the control group and revealed the highest value in the group with PDR. However, analysis of the blood flow and blood velocity showed no statistically significant differences between the diabetic and control groups [39]. AO is a superior technique in comparison with other modern imaging techniques essentially because it reveals pathological changes at a very early stage.

In Ueno’s study, the patients were unaware of the changes in the eye caused by diabetes. All the more interesting is thus a study where the applicability of AO was proven in a group of people in a pre-diabetic state. In total, 34 people with pre-diabetes were included in a comparative study, after first excluding diabetes, cardiovascular diseases, brain diseases, or severe dyslipidaemia. There was a statistically significantly higher WLR (0.29 ± 0.05 vs. 0.22 ± 0.04; *p* < 0.001) and a smaller lumen diameter (LD 94.3 ± 10.9 vs. 105.6 ± 14.6 *p* = 0.022) in the pre-diabetic group compared to the control group. WCSA was also higher in the pre-diabetic group (4519 ± 905 vs. 4284 ± 982; *p* = 0.490), but statistical significance was not established, possibly due to the small size of the study group. In contrast to the current study, a positive correlation was found between WLR, WCSA, and the level of total cholesterol. Similarly to the findings from the current study, multivariate regression analysis showed a positive correlation between WLR and BMI [40].

The same authors studied a group of 47 women with a mean age of 48.9 ± 8.1 years, including 28 overweight and obese women with a mean BMI of 31.1 ± 3.8, in whom diabetes, pre-diabetes, and hypertension were excluded. The control group consisted of 19 women with a BMI within the normal range and a mean value of 23.8 ± 1.2. There were statistically significantly higher WLR (90.25 ± 0.03 vs. 0.29 ± 0.03) and WCSA (4136.7 ± 1140.0 vs. 5217.3 ± 944.0) in the group with an elevated BMI compared to the control group (*p* < 0.001) [41]. The above results and findings of the current study are correspondent to those of Meixner, who also showed a positive correlation between the retinal vascular WLR and BMI. Furthermore, he discovered a positive correlation of WLR with age and hypertension [42]. The impact of hypertension on retinal vascular remodelling in diabetic patients was the subject of further studies with AO. With an rtx1 adaptive optics retinal camera, also used in the current study, Koch studied 49 subjects without DM with a mean age of 44.9 ± 14.4 years, 19 of whom had hypertension (HTN), and 30 did not. Hypertensive subjects presented a mean SBP 154 ± 14 and DBP 95.5 ± 10 mmHg, significantly higher than in the group with DM1 in the current study. In the group of HTN patients compared to the normotensive group, a statistically significantly higher WLR of 0.36 ± 0.08 vs. 0.285 ± 0.05 (*p* < 0.01), respectively, was demonstrated. Moreover, WLR correlated positively with age and mean blood pressure. WCSA was also assessed but, as in the current study, no statistically significant differences were obtained. Researchers have demonstrated the usefulness of AO in the diagnosis of hypertensive retinopathy [43]. This is of utmost importance because HTN occurs twice as often in the DM population as in the non-DM population [44]. Other data indicate that HTN occurs in up to 75% of DM patients, and that the co-occurrence of HTN and DM promotes the progression of microvascular and macrovascular complications [45,46,47]. In the current study, the mean systolic pressure in the DM1 group was 135 ± 14 mmHg and the mean diastolic pressure was 85 ± 9 mmHg, and these values did not correlate with vascular changes, which supports the need to maintain normotension in the patients. In another study, Rosenbaum used an rtx1 camera to discover a positive correlation of retinal arteriolar WLR with blood pressure values. In his study, both elevated blood pressure values and age independently contributed to increased WLR as a result of thickening of the wall of the analysed vessels. Hypotensive treatment normalised WLR and prevented microvascular changes [48]. Additinoally, Gallo proved that there is an impact of HTN on retinal microvascular changes. His study included 1500 subjects aged 55.3 ± 13.0, of whom 324 had DM and 276 had HTN. Statistically significantly higher WLR was observed in the HTN group, while WT and WCSA were not different between the two groups. Unfortunately, patients with diabetes were not selected for further research with AO [49]. Another paper focusing on the effect of hypertension on retinal vascular remodelling was presented by Arichika et al. The analysed group included 51 healthy subjects and 22 HTN patients (without DM), aged >50 years with normal BMI. They found significantly higher WLR in the hypertensive group vs. the control group, 0.280 ± 0.040 vs. 0.315 ± 0.066, respectively (*p* = 0.04). In that study, the mean systolic blood pressure was 140.0 ± 16.9 mmHg, whereas in the current study in the DM1 group it was 135 ± 14 mmHg, which seems to be a safe value rather than a risk factor for vascular remodelling. The same Japanese study showed a strong positive correlation between age (mean 59 ± 4) and WLR (*p* < 0.0001), but the current study involved much younger patients, with a mean age of 43 ± 9, and with the age, gender, and BMI adjustment WLR correlated positively with the duration of diabetes [50]. Interesting results of studies comparing research methods were presented by Nardin, who emphasised the role of AO as a non-invasive research method which is relatively easy to conduct. She noted that AO has good reproducibility and a strong linear correlation between WLR of retinal arterioles and ratio of the tunica media thickness to internal lumen (M\L ratio) of subcutaneous small optic arterioles. Micro-myography is an invasive technique, more difficult to perform, so AO has an advantage here, but it is still not very commonly used and needs further research and validation [51]. Akyol voices a similar opinion in his publication from 2021, where he confirms wide applicability of AO, presenting opportunities but also limitations of this still new diagnostic method [16].

The main limiting factor in examining the structure of the retinal vessels of the eye with the use of rtx1 is the transparency of the eye. Diabetes mellitus is often accompanied by cataracts or/and glaucoma, which may affect the retinal vascular changes; therefore, only patients without the aforementioned diseases were selected for the study. The limitation of our research includes the size and diversity of the studied cohorts. Planning future studies, we will focus on increasing the study group, which will be divided into subgroups depending on the DR severity stage. Furthermore, we aim to reassess the retinal vessels with AO at various time intervals.

The detection of changes in the retinal vessels at such an early stage obliges the physician to educate the patient and intensify the treatment of risk factors for the development of DR. These efforts include improvement of glycaemic and lipid parameters, as well as blood pressure control. Although to date there is no specific treatment for diabetic eye disease diagnosed at such an early stage, the detection of the presented changes should result in regular ophthalmic supervision to prevent the development of full-blown DR. We propose that patients with the characteristic vascular changes in AO, even in the case of normal eye fundus examination, should be treated as patients with already existing mild or moderate NPDR and undergo ophthalmological examinations every 6–12 months according to the recommendations of the American Diabetes Association (ADA) and the European Association for the Study of Diabetes (EASD) [5,52].

## 5. Conclusions

The current study on the use of the AO technique demonstrated the presence of significant morphological changes in the retina of patients with DM1 treated according to current standards without previously detected DR in routine ophthalmological examinations. Referring to our own studies and a review of the available literature, it can be said that WLR is the most accurate marker confirming retinal vascular remodelling. It is hoped that AO will be more widely available and become a routine tool for evaluation of the eye in diabetic patients, especially in patients with a long duration of diabetes without previously detected DR. The current study shows that researchers are obliged to search for further risk factors for the occurrence and progression of vascular lesions, and their work is likely to contribute to better prevention and treatment of DR.

## Figures and Tables

**Figure 1 biomedicines-10-01926-f001:**
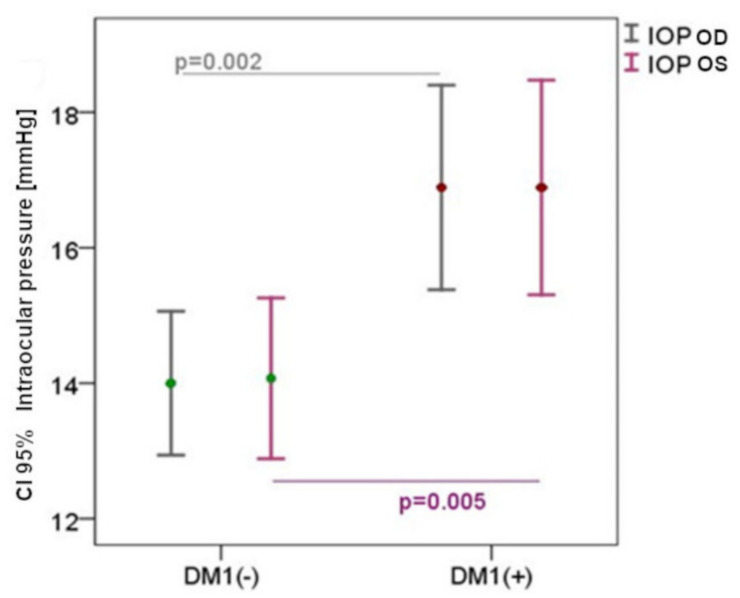
Left eye intraocular pressure (IOPos) and right eye intraocular pressure (IOPod) values in the DM1 group (DM1+) and the control group (DM1−), confidence interval (CI).

**Figure 2 biomedicines-10-01926-f002:**
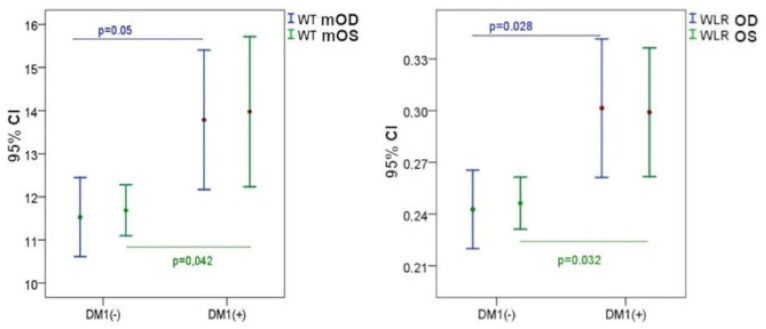
WT and WLR values in the group of DM1 patients (DM1+) and the control group (DM1−), left eye (OS), right eye (OD), and confidence interval (CI).

**Figure 3 biomedicines-10-01926-f003:**
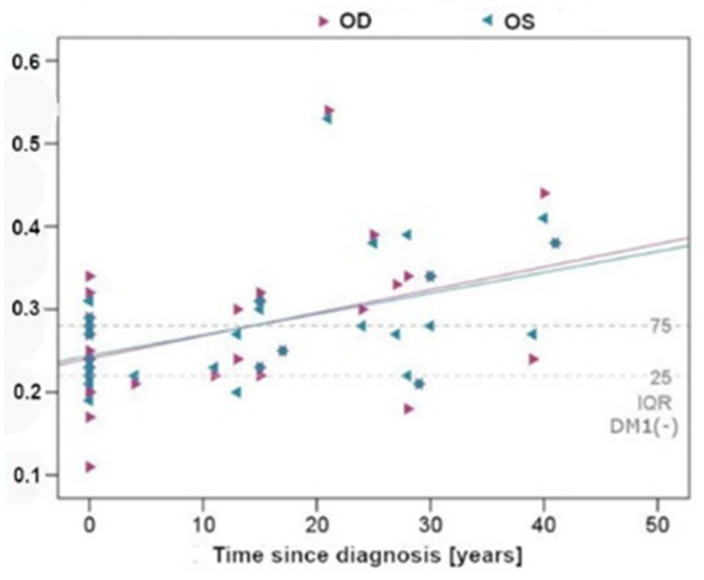
Correlation between WLR and DM1 duration. IQR (interquartile range 25 percentile–75 percentile) for the control group (duration of diabetes is zero) is indicated in the graph, left eye (OS), and right eye (OD), Asterisk same values in different patients.

**Table 1 biomedicines-10-01926-t001:** Normal and recommended ranges of the assessed markers of metabolic balance in diabetes.

Assessed Laboratory Markers	Recommended Range
Glycated haemoglobin (HbA1c)	≤7.0% (≤53 mmol/mol)
Concentration of total cholesterol	<200 mg/dL (<5.2 mmol/L)
Concentration of HDL	>40 mg/dL (>1.0 mmol/L) in men >45 mg/dL (>1.2 mmol/L) in women
Concentration of LDL	<100 mg/dL (2.6 mmol/L)
Concentration of triglycerides	<150 mg/dL (<1.7 mmol/L)
Concentration of creatinine	0.6–1.3 mg/dL (53–115 µmol/L)
Glomerular filtration rate (GFR)	<90 mL/min/1.73 m^2^
Albumin/creatinine ratio (ACR)	<2.5 mg/g creatinine

**Table 2 biomedicines-10-01926-t002:** Characteristics of the study group with metabolic control markers. DDY—diabetes duration years; TG—triglycerides; ACR—albumin to creatinine ratio; BP—blood pressure; BMI—body mass index; IQR—interquartile range (25 percentile; 75 percentile).

	Study Group		
	DM1	Control Group	*p*
**n (%)**	22 (49)	23 (51)	-
**Age, years (SD) mean ± SD**	43 ± 9	41 ± 11	0.54
**Median (IQR)**	41 (37; 51)	42 (33–48)	
**DDY mean ± SD**	22.6 ± 10.0	-	-
**Median (IQR)**	22.5 (15–29.25)		
**HbA1c (%) mean ± SD**	7.5 ± 1.1	5.26 ± 0.18	0.04
**Median (IQR)**	7.45 (6.8–8.7)	5.27 (5.11–5.45)	
**Total cholesterol (mg/dL) mean ± SD**	196 ± 46	185.2 ± 23.4	0.54
**Median (IQR)**	191 (158–232)	184 (152–222)	
**HDL cholesterol (mg/dL) mean ± SD**	77 ± 21	63.0 ± 9.76	0.27
**Median (IQR)**	73 (63–94)	61 (55–85)	
**LDL cholesterol (mg/dL) mean ± SD**	127 ± 42	116.67 ± 31.76	0.05
**Median (IQR)**	114 (93–166)	117 (72–174)	
**TG (mg/dL) mean ± SD**	86 ± 38	101.67 ± 33.13	0.71
**Median (IQR)**	76 (55–115)	92 (54–141)	
**Creatinine (mg/dL) mean ± SD**	0.86 ± 0.13 0.85 (0.80–0.90)	0.93 ± 0.18	0.27
**Median (IQR)**		0.88 (0.76–0.90)	
**eGFR (mL/(min/1.73m^2^)) mean ± SD**	83.8 ± 13.4 82 (77–93)	80.21 ± 11.05	0.99
**Median (IQR)**		81 (78–92)	
**ACR (mg/mmol) mean ± SD**	1.28 ± 1.72 0.45 (0.20–1.95)	0.61 ± 0.69	0.69
**Median (IQR)**		0.59 (0.11–1.25).	
**Systolic BP (mHg)**	135 ± 14 135 (125; 147)	128 ± 9	0.053
		127 (123; 136)	
**Diastolic BP (mmHg)**	85 ± 9 83 (79; 94)	84 ± 9	0.71
		85 (78; 89)	
**BMI [kg/m^2^] mean ± SD**	24.4 ± 2.6 24.0 (22.0–26.4)	23.7 ± 3.0	0.27
**Median (IQR)**		22.3 (21.6–26.0)	
**Waist circumference (cm) mean ± SD**	80.73 ± 11.4 80.5 (70.5–87.8)	81.87 ± 11.81	0.34
**Median (IQR)**		78 (71–91)	

**Table 3 biomedicines-10-01926-t003:** WT, WLR, and WCSA values in the DM1 group (DM1+) and the control group (DM1−); left eye (l), right eye (r).

	DM1 Control Group (−)		DM1 Patients Group (+)		*p*
	Mean ± SD	Median (IQR)	Mean ± SD	Median (IQR)	
WT_r (μm)	11.5 ± 2.1	11.5 (10.2; 12.3)	13.6 ± 3.5	12.4 (11.0; 16.0)	0.050
WT_l (μm)	11.7 ± 1.3	11.5 (10.7; 12.2)	13.9 ± 3.7	13.7 (11.0; 16.3)	0.042
WLR_r	0.24 ± 0.5	0.24 (0.22; 0.28)	0.30 ± 0.09	0.30 (0.23; 0.34)	0.028
WLR_l	0.25 ± 0.03	0.24 (0.22; 0.28)	0.30 ± 0.08	0.28 (0.23; 0.36)	0.032
WCSA_r (μm^2^)	3893 ± 967	3731(3208; 4667)	4661 ± 1825	4470 (3532; 5560)	0.098
WCSA_l (μm^2^)	3836 ± 878	3882 (3412; 4232)	4837 ± 2018	4353 (3577; 5952)	0.089

**Table 4 biomedicines-10-01926-t004:** Multivariate analysis of correlation of WT, WLR, and WCSA with duration of diabetes in patients with DM1, left eye (l), and right eye (r).

Partial Correlation Controlled by: Age, Gender, BMI	Time from DM1 Diagnosis	
	r	*p*
WTr (μm)	0.452	0.004
WTl (μm)	0.489	0.001
WLRr	0.454	0.004
WLRl	0.477	0.002
WCSAr (μm^2^)	0.322	0.046
WCSAl (μm^2^)	0.398	0.012

## Data Availability

The data that support the findings of this study are available from the corresponding author, upon reasonable request.

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
