# Peer review of "Evaluation of Morphological Changes in Retinal Vessels in Type 1 Diabetes Mellitus Patients with the Use of Adaptive Optics"

_biomedicines, 2022, doi:10.3390/biomedicines10081926_

Round 1

Reviewer 1 Report

Overall, this is well-written manuscript on the use of the adaptive optics (AO) technique demonstrating the presence of significant morphological changes in the retina of patients with diabetes mellitus type 1 (DM1) treated according to current standards without previously detected DR in routine ophthalmological examinations. The DM1 group had significantly higher the vessel wall thickness, wall-to-lumen ratio and vascular wall cross-sectional area. These parameters correlated significantly with the duration of diabetes, but not with intraocular pressure.

Author Response

Dear Sir or Madame,

Thank you for the insightful and thorough review of our work. We appreciate your effort involved in reviewing such an extensive text. We considered all your comments and we have provided the text for a linguistic expertise, we hope that it will positively affect the quality of our text. We wish the revised manuscript now meets your and your readers' expectations.

Yours faithfully,

Authors

Please see the attachment -revised manuscript (also with corrections suggested by a second reviewer)

Reviewer 2 Report

The study projects the use of adaptive optics to detect early events in DR in DM1 patients with no DR reported by Fundus imaging. This is a technique in the development and may have the potential for future use. Currently, there are several concerns that need to be addressed prior to its application. Hence, the authors should consider including these limitations in the study and their potential to develop for future use. A few limitations/concerns/suggestions are below:

Is it specific to DM1? What makes the vascular pathologies different in DM2 versus DM1, or are these the same? This must be discussed in the manuscript.

Even if it is detected early, what would be the use of this method in treating the patient? Are there any treatments developed for patients with DR in such an early stage? What control measures can be adopted to manage such patients? Please include a discussion in the manuscript, so the readers can assess the value of the findings.

Table 3 indicates statistically significant changes in WT, WLR, and WCSA. Are they clinically significant- the variability between the samples, particularly DM1 patients is very high. Does that mean that the patients are not in the same stage of the disease, or is it the lack of reliability that the vascular changes cannot be detected in all DM1 patients- E.g. the Control patients have 12.3 on certain occasions and there are DM1 patients with a score of 11 in WT analyses? Please comment on the reliability of the technique based on the data.

Please detail the study limitations in general and how future studies can be designed to overcome the limitations and improve the data.

Author Response

Dear Sir or Madame,

Thank you for the insightful and thorough review of our work. We appreciate your effort involved in reviewing such an extensive text. We considered all your comments and we hope to have resolved all the issues you suggested and correct the mistakes.

Below are the answers to your questions:

  1.  

The changes we have observed in DM1-patients are not specific to DM1, as they can also be found in DM2 and other types of diabetes. The results of research carried by other authors cited in our discussion also revealed the presence of similar vascular lesions, although they analyzed patients with obesity, pre-diabetes and diabetes type 2. With the current state of knowledge, however, we cannot rule out the occurrence of significant differences in the image of damaged vessels in the aforementioned groups of patients. Thus, further research is required in this area. The development of the described vascular changes is the effect of hyperglycemia, which is common to all types of diabetes. In subsequent studies, we are planning to analyze the impact of arterial blood pressure on the image of retinal vessels in AO.

  1.  

The detection of changes in the retinal vessels at such an early stage obliges the physician to educate the patient and intensify the treatment of risk factors for the development of DR. These efforts include improvement of glycaemic and lipid parameters, as well as blood pressure control. Although to date there is no specific treatment for diabetic eye disease diagnosed at such an early stage, the detection of the presented changes should result in regular ophthalmic supervision to prevent the development of full-blown DR. We propose that patients with the characteristic vascular changes in AO, even in case of normal eye fundus examination, should be treated as patients with already existing mild or moderate NPDR and undergo ophthalmological examinations every 6-12 months according to the recommendations of the American Diabetes Association (ADA) and the European Association for the Study of Diabetes (EASD).

  1. (Table 3).

Thank you very much for this important remark. Unpublished reliability studies of morphological parameters (vessel diameter (VD), lumen diameter (LD), Wall CrossSection Area (WCSA), Wall Thickness (WT), and Wall to Lumen Ratio (WLR) ) of retinal arterioles measured by adaptive optics  were conducted before any projects to study retinal vascular morphology in patients with diabetes, hypertension and healthy control groups. The results of these studies were consistent with the studies by Koch et al. (1). The measurements were performed in the same conditions as in this study. To assess reliability the coefficient of variation (CV=Standard Deviation *100% / Mean) and α-Cronbach coefficient (α-CC) of internal correlation were calculated for 3 models from images of 5 patients (2 hypertensive, 1 diabetes, 2 control; 3 male; mean age 42 ± 26 years):

  1. Intraobserver reliability: obtained images were analyzed twice by the same observer at least one week apart.
  2. Interobserver reliability: obtained images were analyzed individually by two observers (a novice student, and professor with over 25 years of experience in retina microcirculation study).
  3. Test-retest reliability of the method: obtained 3 images from the 5 patients measured during one visit in two minutes to reduce the biological and the environmental variabilities were analyzed by one observer.

All evaluated parameters proved to be very good reliable with CV< 5% and α-CC> 0,88 for all measured parameters of arteriolar structures by rtx1.

SD- standard deviation of the studied mean values ​​of these parameters may be high due to :

  1. The diameters of the examined retinal vessels differ significantly between patients (even by 100%). Thus, also the thickness of the retinal walls differ, as this parameter correlates with the diameter of the vessels;
  2. The studied group of patients varied in terms of duration of DM, diet and lifestyle. These discrepancies may explain the differences in vascular changes in this group.

  1. Koch E, Rosenbaum D, Brolly A, Sahel JA, Chaumet-Riffaud P, Girerd X, Rossant F, Paques M. Morphometric analysis of small arteries in the human retina using adaptive optics imaging: relationship with blood pressure and focal vascular changes. J Hypertens. 2014 Apr;32(4):890-8. doi: 10.1097/HJH.0000000000000095. PMID: 24406779; PMCID: PMC3966915.

Limitation:

The main limiting factor in examining the structure of the retinal vessels of the eye with the use of rtx1 is the transparency of the eye. Diabetes mellitus is often accompanied by cataracts or/and glaucoma, which may affect the retinal vascular changes, therefore only patients without the aforementioned diseases were selected for the study. In all studies assessing the relationship between retinal vascular system changes and systemic diseases, the most common eye diseases, such as cataracts and glaucoma, should always be excluded.

In this study, only the duration of diabetes was selected as a risk factor for vascular changes. However, other clinical factors, in example blood pressure, may affect the course of vascular changes both in DM - patients and in the control group. Our aim is to conduct such research in the future. Noteworthy, in this work we investigated changes in the microvascular system. However, it would be interesting to assess similar vascular parameters in large vessels (e.g. non-invasive Intima-Media Thickness Lumen Ratio of the carotid artery and comparison of these changes in the retina of the eye) in the same patients, which will also be the subject of our future research.

The limitation of our research include the size and diversity of the studied cohorts. Planning future studies, we will focus on increasing the study group, which will be divided into subgroups depending on the DR severity stage. Furthermore, we aim to reassess the retinal vessels with AO at various time intervals. 

Moreover, we changed the conclusions to follow directly from the conducted research results. Also we have provided the text for a linguistic expertise, we hope that it will positively affect the quality of our text. We wish the revised manuscript now meets your and your readers' expectations.

Yours faithfully,

Authors

Round 2

Reviewer 2 Report

Authors have addressed my concerns